# Structure and inhibition mechanism of the catalytic domain of human squalene epoxidase

Anil K. Padyana [1], Stefan Gross [1], Lei Jin[2], Giovanni Cianchetta[1,5], Rohini Narayanaswamy[1], Feng Wang[3], Rui Wang[3,6], Cheng Fang[4], Xiaobing Lv[4,7], Scott A. Biller[1], Lenny Dang[1], Christopher E. Mahoney [1], Nelamangala Nagaraja[1], David Pirman[1], Zhihua Sui[1], Janeta Popovici-Muller[1,8] & Gromoslaw A. Smolen [1,9]

Squalene epoxidase (SQLE), also known as squalene monooxygenase, catalyzes the stereospecific conversion of squalene to 2,3(S)-oxidosqualene, a key step in cholesterol biosynthesis. SQLE inhibition is targeted for the treatment of hypercholesteremia, cancer, and fungal infections. However, lack of structure-function understanding has hindered further progression of its inhibitors. We have determined the first three-dimensional high-resolution crystal structures of human SQLE catalytic domain with small molecule inhibitors (2.3 Å and 2.5 Å). Comparison with its unliganded state (3.0 Å) reveals conformational rearrangements upon inhibitor binding, thus allowing deeper interpretation of known structure-activity relationships. We use the human SQLE structure to further understand the specificity of terbinafine, an approved agent targeting fungal SQLE, and to provide the structural insights into terbinafine-resistant mutants encountered in the clinic. Collectively, these findings elucidate the structural basis for the specificity of the epoxidation reaction catalyzed by SQLE and enable further rational development of next-generation inhibitors.

[1] Agios Pharmaceuticals, 88 Sidney Street, Cambridge, MA 02139, USA. [2] Agile Biostructure Solutions Consulting, LLC, 8 Harris Ave, Wellesley, MA 02481, USA. [3] Wuxi Biortus Biosciences Co. Ltd., 6 Dongsheng West Road, Jiangyin 214437, China. [4] Shanghai ChemPartner Co. Ltd., 998 Halei Road, 201203 Shanghai, China. [5] Present address: KSQ Therapeutics, 610 Main St, Cambridge, MA 02139, USA. [6] Present address: Department of Stomatology, Xiamen University, 361102 Xiamen, China. [7] Present address: Sundia MediTech Company, Ltd., 917 Halei Road, 201203 Shanghai, China. [8] Present address: Decibel Therapeutics, 1325 Boylston St Suite 500, Boston, MA 02215, USA. [9] Present address: Celsius Therapeutics, 215 First Street, Cambridge, MA 02142, USA. Correspondence and requests for materials should be addressed to Anil K. Padyana (email: anil.padyana@agios.com)

The interest in cholesterol has been primarily fueled by the connection of hypercholesterolemia to coronary heart disease[1]. Statins, inhibitors of the rate controlling biosynthetic step HMG-CoA reductase (HMGCR), have transformed this disease landscape and motivated extensive efforts to identify additional agents capable of modulating cholesterol homeostasis[2]. In turn, the availability of these inhibitors has facilitated the identification of additional disease settings where cholesterol pathway modulation may potentially be used for therapeutic purposes. For example, the widespread use of statins has led to epidemiological observations of lowered cancer incidence[3]. However, the efficacy of cholesterol pathway inhibitors in oncology treatment regiments remains to be demonstrated and progress is likely hindered by the lack of clear patient selection strategies. Interestingly, we recently identified a subset of neuroendocrine tumors with unexpected sensitivity to inhibition of SQLE, a step in the cholesterol biosynthetic pathway[4].

SQLE is a flavin adenosine dinucleotide (FAD)–dependent epoxidase that catalyzes stereospecific conversion of non-sterol intermediate squalene to 2,3(S)-oxidosqualene (Fig. 1a), the first oxygenation step in cholesterol synthesis. SQLE is proposed to be the second rate-limiting enzyme of the cholesterol biosynthesis downstream of HMGCR[5,6] and, as such, is regulated on multiple levels. First, SQLE is a direct target of SREBP2 transcription factor that regulates the majority of genes in the cholesterol biosynthetic pathway[6]. Second, the N-terminus of SQLE protein appears to contain a cholesterol sensing domain that regulates proteasomal degradation of SQLE in a cholesterol-dependent manner[7] by relying on MARCH6, an E3 ubiquitin ligase[8]. Interestingly, unsaturated fatty acids, such as oleate, can stabilize SQLE by interfering with MARCH6-mediated degradation[9]. Collectively, these studies highlight multiple mechanisms regulating SQLE to allow for finely-tuned cholesterol homeostasis. In fungi, SQLE is involved in ergosterol biosynthesis and its inhibitor, terbinafine, is approved for the treatment of specific fungal infections[10].

Based on a growing understanding of the structural and catalytic properties of FAD monooxygenases, SQLE has been classified as a Group E monoxygenase that requires external NADPH-cytochrome P450 reductase (P450R) as an electron donor for squalene epoxidation[11]. Small molecule inhibitors of human SQLE, such as NB-598 and compound-4′′ (Cmpd-4′′), have been previously reported with IC$_{50}$ in the range of 10–60 nM (Fig. 1b)[12,13]. However, further improvements of these compounds have been hampered by the lack of structural knowledge and by the absence of detailed understanding of their inhibition mechanism.

Here we report the high-resolution crystal structures of the human SQLE catalytic domain and identify critical conformational rearrangements necessary for inhibitor binding. We use the human SQLE structure to further understand the specificity of terbinafine, an approved therapeutic targeting fungal SQLE, and provide the structural insights into terbinafine-resistant mutants encountered in the clinic. Lastly, we model the SQLE substrate, squalene, into the enzyme active site and elucidate the structural basis for the specificity of the catalyzed epoxidation reaction. Collectively, this work provides a foundation for further development of the next generation of SQLE inhibitors.

## Results

### Functional characterization of human SQLE. 
We developed a robust method for expression and purification of recombinant human SQLE proteins for structural and biophysical studies by truncating N- or C-terminal membrane regions (Methods). Individual recombinant proteins were assessed using a thermal shift assay to evaluate their stability and selection of inhibitors for the crystallography experiments (Supplementary Fig. 1). We observed significant stabilization upon addition of NB-598 and Cmpd-4′′ inhibitors ($\Delta T_m$ 19.7 °C and 17.6 °C, respectively) to the N-terminally truncated SQLE (118–574), resulting in the selection of this construct for further studies.

To confirm that the construct used in crystallography is enzymatically competent, we compared SQLE biochemical activity in multiple systems and contexts. First, we used the recombinant N-terminally truncated SQLE (118–574). Second, we overexpressed full-length SQLE in Sf9 cells using a baculovirus system and utilized a membrane preparation termed baculosomes[14], analogous to microsomes, as the source of SQLE protein. Finally, we used human liver microsomes (HLM) which have the advantage of providing endogenous SQLE, but contain a full complement of drug metabolizing P450s that may confound the analysis of inhibitor effects. We also developed a liquid chromatography–mass spectrometry (LC-MS) method to directly measure the product of the SQLE reaction, 2,3-oxidosqualene, which provided increased throughput over previously described thin layer chromatography-based assay system[15]. We compared the activity of SQLE (118–574) with that of full-length SQLE in baculosome preparations and the endogenous SQLE from HLM, and found that affinities for both FAD (5.2 ± 0.5 μM for SQLE (118–574), 8.1 ± 0.6 μM for baculosome SQLE, 9.6 ± 0.5 μM for HLM), and squalene (1.9 ± 0.4 μM for SQLE (118–574), 3.3 ± 0.7 μM for baculosome SQLE, 2.9 ± 0.2 μM for HLM) did not differ significantly among the three systems (Fig. 1c and Table 1). The $k_{cat}$ and $k_{cat}/K_M$ for SQLE (118–574) were 2.09 ± 0.12 min$^{-1}$ and 1.10 ± 0.9 × 10$^6$ M$^{-1}$ min$^{-1}$, for baculosome SQLE were 1.40 ± 0.3 min$^{-1}$ and 4.25 ± 0.8 × 10$^5$ M$^{-1}$ min$^{-1}$, respectively. The $k_{cat}$ and $k_{cat}/K_M$ for the HLMs were approximately ten-fold lower at 0.21 ± 0.02 min$^{-1}$ and 1.38 ± 0.4 × 10$^4$ M$^{-1}$ min$^{-1}$, possibly due to loss of specific activity in the process of microsome preparation. The kinetic parameters obtained in these studies confirm the robustness of the recombinant proteins and place them well within the range of other metabolic enzymes[16].

### SQLE structure in complex with inhibitors. 
We determined the structure of human SQLE de novo by crystallizing the ternary complex of L-selenomethionine (SeMet)-incorporated protein with Cmpd-4′′, FAD and by performing multiwavelength anomalous dispersion experiments to derive the phases used for structure solution (Table 2). The initial model was improved by extensive manual rebuilding and by refining against the higher resolution (2.30 Å) native dataset. The final model consists of SQLE amino acid residues 121–574 with well-defined electron density for the FAD and Cmpd-4′′ (SQLE•FAD•Cmpd-4′′) (Supplementary Fig. 2a). Using the same crystallization conditions, we also determined the NB-598 complex structure (SQLE•FAD•NB-598) at 2.50 Å resolution using phases derived from the SQLE•FAD•Cmpd-4′′ complex (Supplementary Fig. 2b). The overall structure of SQLE exhibits the predicted split domain architecture with FAD and substrate-binding domains interspersed within the primary structure, followed by helical membrane-binding domain at the C-terminus (Fig. 2a, Supplementary Fig. 3). The FAD-binding domain adopts a three-layer ββα sandwich architecture, using the CATH nomenclature[17], which is alternatively referred to as the GR2 Rossman fold[11]. The substrate-binding domain adopts a two-layer βα sandwich domain with seven-stranded β-sheet structure, followed by two helices at the C-terminus. While we observed a dimeric SQLE molecule in the asymmetric unit, we speculate that this is a technical artifact, since characterization by size-exclusion chromatography showed it to be a monomer in solution (Supplementary Fig. 4).

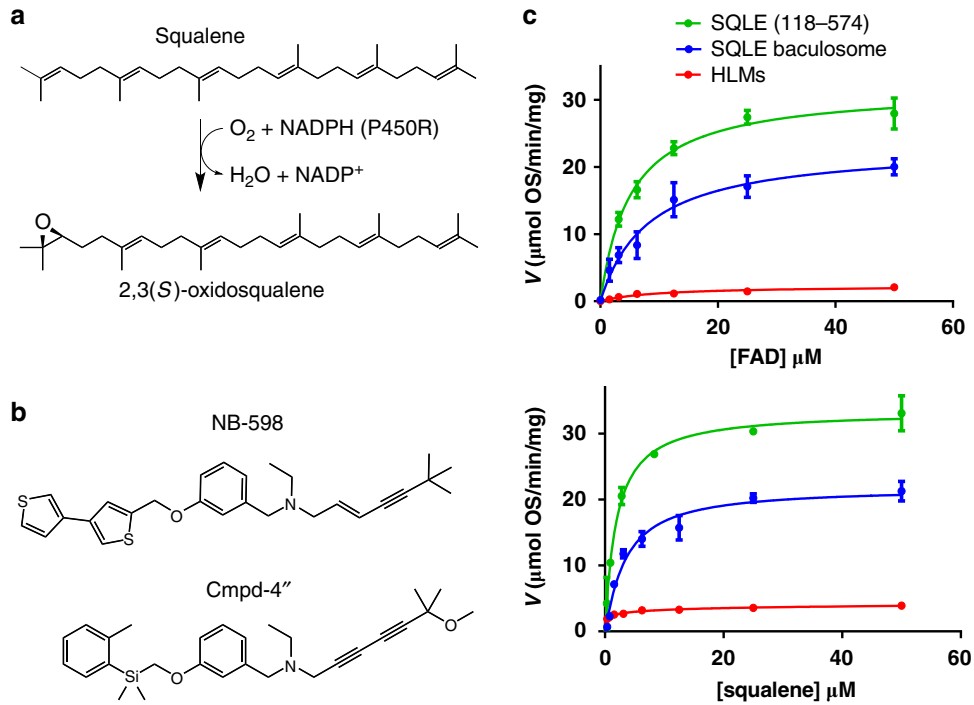

**Fig. 1** Structure of SQLE inhibitors and biochemical characterization of SQLE reaction. **a** Biochemical reaction catalyzed by SQLE. **b** Structures of SQLE inhibitors NB-598 and Cmpd-4″. **c** Steady-state parameters for squalene and FAD affinity. Three experimental systems utilized correspond to the recombinant N-terminally truncated SQLE (118–574), baculosome membrane preparations from Sf9 cells overexpressing full-length SQLE, and human liver microsomes (HLMs) with endogenous SQLE. Enzyme concentrations were determined by immunoblotting and comparing to a standard curve of recombinant purified SQLE. Data obtained from enzymatic assays performed with saturating squalene and varied FAD or varied squalene and saturating FAD were fit to the standard Michaelis–Menten model to obtain $K_M$ values for each varied reaction component. Points and error bars denote the mean and standard deviation of triplicate measurements, respectively

**Table 1 Kinetic parameters for the SQLE reaction in different experimental models**

|  | FAD $K_M$ (μM) | Squalene $K_M$ (μM) | $k_{cat}$ (min$^{-1}$) | $k_{cat}/K_M$ (M$^{-1}$ min$^{-1}$) |
|---|---|---|---|---|
| SQLE (118–574) | 5.2 ± 0.5 | 1.9 ± 0.4 | 2.09 ± 0.12 | 1.10 ± 0.9 × 10$^6$ |
| SQLE baculosome | 8.1 ± 0.6 | 3.3 ± 0.7 | 1.40 ± 0.3 | 4.25 ± 0.8 × 10$^5$ |
| HLMs | 9.6 ± 0.5 | 2.9 ± 0.2 | 0.21 ± 0.02 | 1.38 ± 0.4 × 10$^4$ |

The structures clearly show that Cmpd-4″ and NB-598 bind in a similarly extended conformation in a common site and are surrounded by primarily non-polar residues (Fig. 2b, Supplementary Fig. 5). This site is deeply buried and is located at the interface of the FAD-binding, substrate-binding, and C-terminal helical domains. No significant conformational change was observed between the two X-ray structures which can be superimposed with a root mean square deviation (RMSD) of 0.14 Å. The two inhibitors share multiple structural features, such as N-benzyl-N-ethanamine linker, the nitrogen of the central linker that is connected to aliphatic chains, and the phenyl ring that is connected to mostly aromatic moieties. The aliphatic groups of both compounds bind deep into a helical bundle region toward C-terminus and are surrounded by L469, L473, C491, F495, P505, L508, L509, L519, H522, F523, and V526. The aromatic groups of the inhibitors occupy the open end of the pocket proximal to FAD and are surrounded by the apolar residues F166, Y195, A322, L333, Y335, P415, L416, and G418. The aromatic rings of the compounds do not seem to establish any stacking interactions with the surrounding aromatic residues. The ethanamine linkers in both X-ray structures bind in an area defined by Y195, I197, I208, L234, L416, T417, L473, F477, F495,

P505, V506, and L509. Both ligands establish a hydrogen bond between the tertiary amine and the hydroxyl moiety of Y195. This hydrogen bond is the only specific and directional interaction established by both compounds with the SQLE. Consistently, the tertiary amine motif is a common feature in all SQLE inhibitors published to date and the interaction with conserved Y195 explains its required presence[18,19].

**Fungal SQLE modeling and terbinafine binding**. Interestingly, terbinafine, an approved agent targeting fungal SQLE, also contains the tertiary amine motif (Fig. 3a). There has been a considerable interest in using terbinafine as a tool to explore the consequences of SQLE inhibition in human cancer cell lines[20,21]. We tested the terbinafine in the HLM assay and determined it to be a weak partial inhibitor with a relative IC$_{50}$ of 7.7 μM and a maximal inhibition of 65% at 100 μM inhibitor concentration (Fig. 3b). This suggests that terbinafine is not optimal for studying human SQLE, particularly when compared to NB-598 or Cmpd-4″. Alignment of SQLE from clinically relevant fungal strains, such as *Trichophyton rubrum*, *Trichophyton mentagrophytes,* and *Candida albicans* to mammalian sequences show that compound binding site is highly conserved (Supplementary Fig. 3a). Three

**Table 2 X-ray data collection, phasing, and refinement statistics**

| | SQLE·FAD· Cmpd-4″ | SeMet-SQLE·FAD·Cmpd-4″ | | | SQLE·FAD· NB-598 | SQLE·FAD |
|---|---|---|---|---|---|---|
| | **Native** | **Peak** | **Inflection** | **Remote** | | |
| | **PDB:6C6N** | | | | **PDB:6C6P** | **PDB:6C6R** |
| **Data collection** | | | | | | |
| Wavelength (Å) | 0.97941 | 0.97909 | 0.97959 | 0.96108 | 0.97915 | 0.9792 |
| Space group | $P3_221$ | $P3_221$ | $P3_221$ | $P3_221$ | $P3_221$ | $P3_221$ |
| Cell dimensions | | | | | | |
| $a$ (Å) | 126.96 | 127.1 | 127.2 | 127.23 | 127.86 | 126.39 |
| $b$ (Å) | 126.36 | 127.1 | 127.2 | 127.23 | 127.86 | 126.39 |
| $c$ (Å) | 166.12 | 165.91 | 166.14 | 166.24 | 165.09 | 166.01 |
| $\alpha, \beta, \gamma$ (°) | 90, 90, 120 | 90, 90, 120 | 90, 90, 120 | 90, 90, 120 | 90, 90, 120 | 90, 90, 120 |
| Resolution[a] (Å) | 35.79–2.30 | 50–2.75 | 50–2.90 | 50–3.15 | 40.57–2.50 | 38.81–3.00 |
| | (2.38–2.30) | (2.80–2.75) | (2.95–2.90) | (3.20–3.15) | (2.59–2.50) | (3.11–3.00) |
| $R_{merge}$[a] (%) | 8.2 (55.5) | 9.0 (53.5) | 8.0 (51.8) | 8.5 (48.9) | 8.6 (54.1) | 12.4 (68.8) |
| $I/\sigma I$[a] | 12.1 (3.0) | 52.1 (7.7) | 39.7 (5.1) | 37.7 (6.4) | 15.1 (2.5) | 10.6 (2.5) |
| Completeness[a] (%) | 98.8 (99.7) | 100 (100) | 100 (100) | 99.9 (100) | 88.5 (89.3) | 99.6 (98.8) |
| Redundancy[a] | 5.2 (5.3) | 18.4 (18.8) | 9.1 (9.4) | 9.1 (9.3) | 4.8 (4.7) | 7.4 (6.1) |
| Figure of merit (post-density modification) | | | 0.24/0.74 | | | |
| $R_{cullis}$ (anomalous) | | | 0.84 | | | |
| **Refinement** | | | | | | |
| No. reflections | 68379 | | | | 48318 | 31143 |
| $R_{work}/R_{free}$ (%) | 18.90/22.00 | | | | 19.31/23.42 | 19.01/24.85 |
| No. of atoms | | | | | | |
| Protein | 7058 | | | | 7050 | 7060 |
| Ligand | 401 | | | | 363 | 315 |
| Water | 179 | | | | 170 | 58 |
| $B$-factors (Å²) | | | | | | |
| Protein | 53.46 | | | | 59.69 | 61.32 |
| Ligand | 57.33 | | | | 61.35 | 57.17 |
| Water | 47.31 | | | | 55.42 | 44.68 |
| R.m.s deviations | | | | | | |
| Bond lengths (Å) | 0.003 | | | | 0.009 | 0.004 |
| Bond angles (°) | 0.58 | | | | 0.99 | 0.68 |

[a]Values in parentheses are for highest-resolution shell

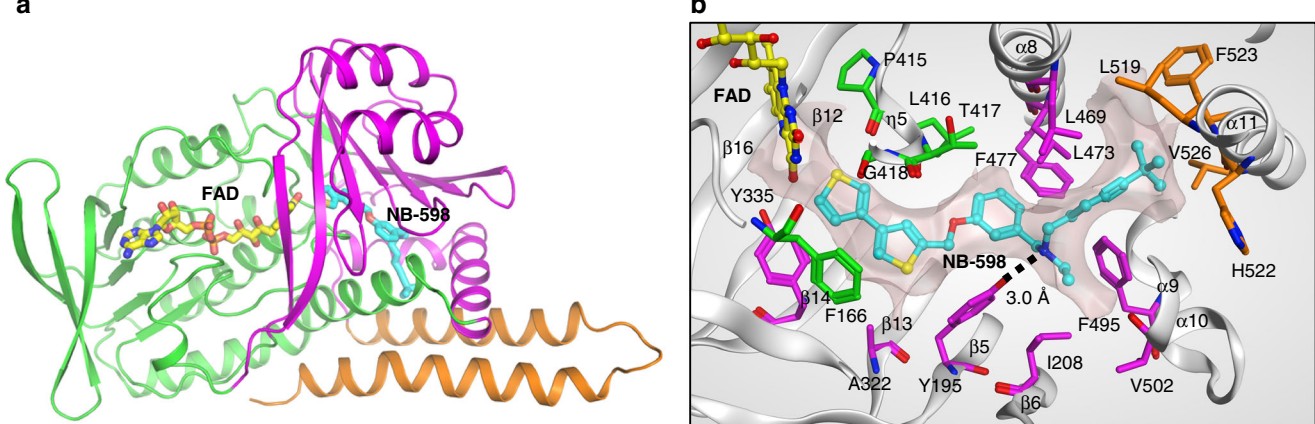

**Fig. 2** Human SQLE structure and inhibition. **a** Overall structure of SQLE bound to inhibitor NB-598. SQLE protein is shown in ribbon representation with the FAD-binding domain in green, the substrate-binding domain in magenta, and the C-terminal membrane–associated helical domain in orange. FAD (yellow) and NB-598 (cyan) are shown as sticks. Hetero-atoms follow the color scheme of yellow, blue, red, and orange for sulfur, nitrogen, oxygen, and phosphor, respectively. **b** NB-598 binding site. NB-598 and FAD are in ball-and-stick presentation with atomic color scheme as described above. The residues forming the compound binding site is in line presentation with the color scheme on carbon atoms to match the domain coloring scheme as in panel **a** with important specific residues shown as sticks. The van der waals (VDW) contact surface of the pocket within the 4.5 Å of NB-598 shown as semi-translucent surface. Hydrogen bond interaction between Y195 and the central amine of NB-598 is shown as black dashed line with the distance labeled

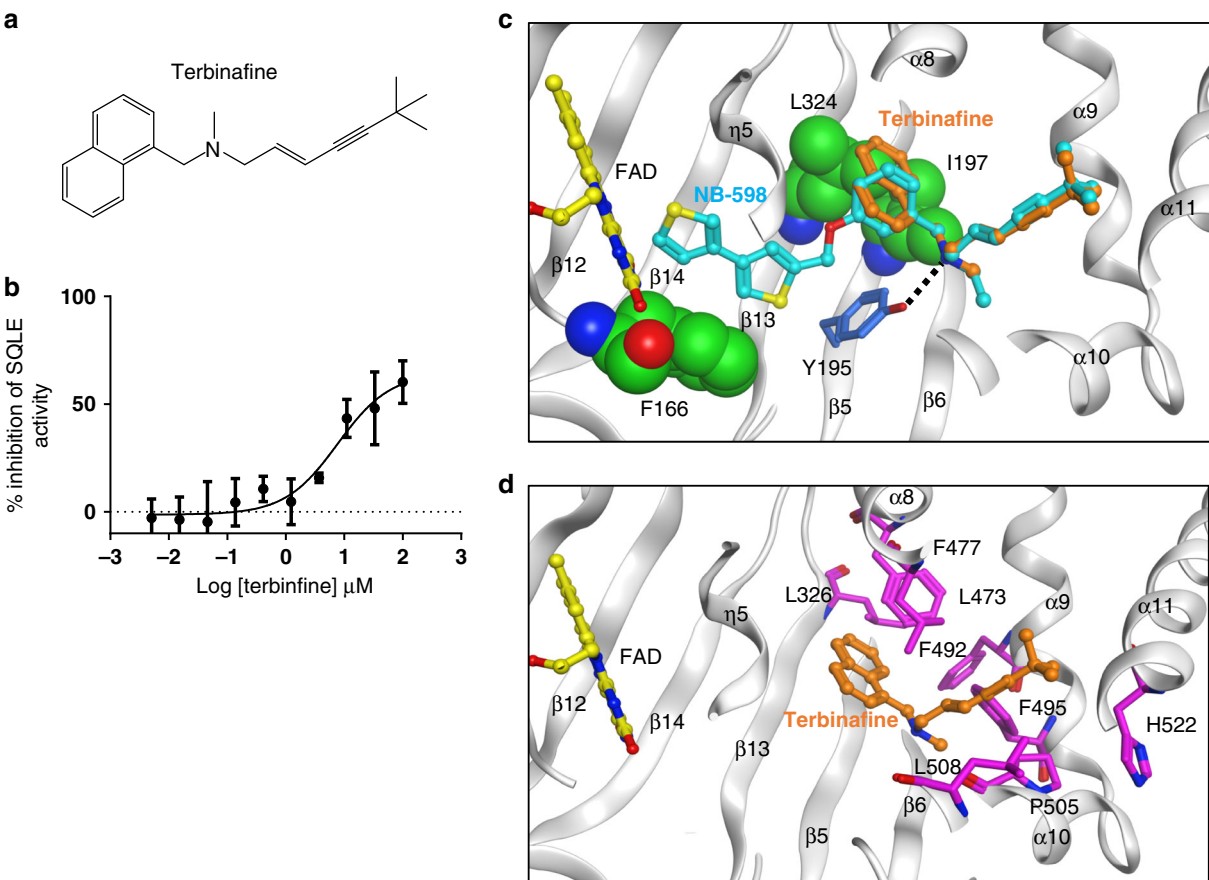

**Fig. 3** Biochemical characterization of terbinafine, binding model and rationale for fungal drug resistance. **a** Structure of terbinafine. **b** Terbinafine is a weak partial inhibitor of human SQLE in the HLM assay. Relative $IC_{50}$ was determined to be 7.7 μM with a maximal inhibition of 65% at 100 μM inhibitor concentration. Error bars represent the standard deviations from a representative experiment performed in triplicate. **c** Superposition of terbinafine structural model with NB-598 using the SQLE•FAD•NB-598 complex. NB-598 (cyan), terbinafine (orange), and FAD (yellow) are shown in ball-and-sticks representation. Non-conserved amino acids in the inhibitor binding site is shown in CPK (green) and the Y195 residue that is conserved across species in stick (blue) representation. Hydrogen bond interaction between Y195 and the central amine moiety of inhibitors is shown as black dashed line. **d** Mapping of terbinafine-resistant mutations to the human SQLE structure with superposed terbinafine model. Equivalent human residues corresponding to terbinafine-resistant mutations identified in fungi are shown as magenta sticks. FAD (yellow) and Terbinafine (orange) are depicted in ball-and-stick representation

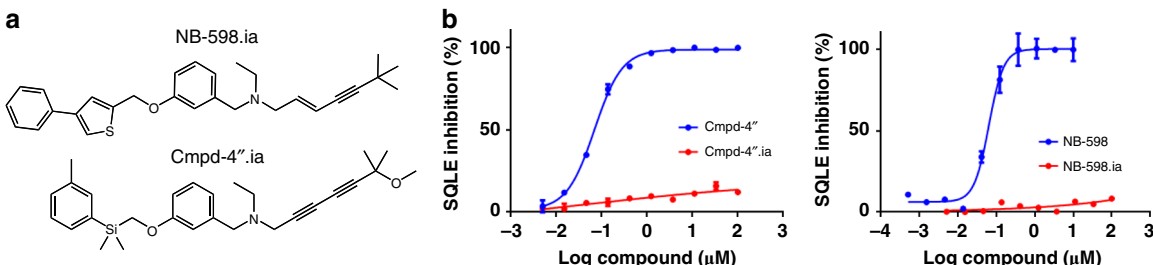

**Fig. 4** Biochemical activity of SQLE inhibitors and their inactive analogs. **a** Structures of inactive analogs, NB-598.ia and Cmpd-4″.ia. **b** NB-598 inhibits baculosome-derived SQLE with an $IC_{50}$ of 63 nM, while the structural analog NB-598.ia has an $IC_{50}$ of >100 μM (left panel) and Cmpd-4″ inhibits baculosome-derived SQLE with an $IC_{50}$ of 69 nM, while the structural analog Cmpd-4″.ia has an $IC_{50}$ of >100 μM (right panel). Points and error bars are the mean and standard deviation of triplicate experiments

amino acids (F166, I197, and L324) positioned near the aromatic side of the inhibitor were not conserved between the species (Fig. 3c), while the amino acids near the linker and the aliphatic side were identical between human and fungal SQLE. The aromatic side of terbinafine contains bulkier naphthalene group in the position of benzene linker of NB-598. Modeling the

terbinafine using NB-598 template in human SQLE positions the naphthalene group adjacent to bulkier hydrophobic side chains of I197 and L324. These sub-optimal non-polar contacts are consistent with the observed higher $IC_{50}$ values of terbinafine in the HLM enzymatic assay. Interestingly, residues corresponding to I197 and L324 in dermatophyte SQLE are smaller hydrophobic

valines, likely resulting in optimal interactions with naphthalene consistent with the reported selectivity profile of terbinafine[10].

Several reports have identified strains resistant to terbinafine treatment with point mutations detected in fungal SQLE (*ERG1* gene) in both clinical and non-clinical settings[22–26]. We mapped the reported resistant point mutations onto the human SQLE sequence and to the SQLE•FAD•NB-598 structure (Fig. 3d, Supplementary Table 1). Remarkably, all the SQLE resistant mutations are in the inhibitor binding pocket. Mutation of these conserved residues in dermophytes (L326, L473, F477, F492, F495, L508, P505, and H522 of human SQLE) would be predicted to affect the non-polar interactions with the inhibitor resulting in the loss of biochemical potency. Collectively, our structural insights provide a detailed explanation for the weak inhibitory potency of terbinafine against human SQLE and offer understanding of the previously identified terbinafine-resistant mutations.

**Design of structurally-related inactive inhibitor analogs**. To further enable cell biology studies and to demonstrate the specificity of the observed cellular responses after the addition of NB-598 or Cmpd-4″, we designed small changes in the compounds to make structurally-related inactive analogs (ia). The narrow binding pocket observed in the inhibitor-bound structures (Fig. 2b and Supplementary Fig. 5) is consistent with the steep structure-activity relationship previously observed[13,27]. We made limited changes to the inhibitor structures (thiophene into benzene in NB-598 and ortho-tolyl into meta-tolyl in Cmpd-4″) on the aromatic side that might result in steric clashes to generate two analogs, termed NB-598.ia and Cmpd-4″.ia (Fig. 4a, Supplementary Fig. 6 and Supplementary Methods). Potent inhibitory activity of NB-598 and Cmpd-4″, and the expected lack of activity of the analogs, was confirmed in the biochemical assay (Fig. 4b).

**Structure of unliganded SQLE and the substrate-binding model**. To elucidate any SQLE conformational changes that may be associated with inhibitor binding, we crystallized and determined the structure of unliganded enzyme (SQLE•FAD) by using phases derived from the SQLE•FAD•Cmpd-4″ complex (Table 2). The overall structure of SQLE•FAD is similar to that of the inhibitor-bound structure (Supplementary Fig. 7). However, the substrate-binding domain exhibits greater displacement (RMSD 0.5 Å) compared to FAD-binding (RMSD 0.3 Å) or the membrane domains (RMSD 0.2 Å) (Fig. 5a). In addition to the overall domain motions, in the unliganded structure the side chain of Y195 adopts a distinct conformation and forms a hydrogen bond with the side-chain amide of Q168 from the FAD binding domain. In this conformation, the two polar side chains are shielded within the non-polar pocket, and the alternate conformation of conserved Y195 is required to form the critical hydrogen bond with the tertiary amine of the inhibitors (Fig. 5a). To further explore the role of Y195 in SQLE catalysis and inhibition, we designed two specific mutations: a conservative Y195F substitution and a more severe Y195A change. Interestingly, both mutations resulted in a > 90% loss of catalytic activity suggesting that hydrogen bonding property of side-chain hydroxyl of Y195 to Q168 side chain is critical for maintaining the SQLE activity (Fig. 5b).

The extended shape and length (~25 Å) of the inhibitor binding pocket with an opening adjacent to flavin group, as well as the predominantly non-polar residues lining the pocket, suggest that it represents the substrate-binding site of SQLE. Our attempts to generate well-diffracting crystals of SQLE•FAD complex with squalene or 2,3(*S*)-oxidosqualene were not successful. Therefore, we modeled squalene binding by conducting a molecular docking experiment on the unliganded SQLE•FAD structure as a template and showed that most of the amino acid residues in the squalene binding pocket overlap with amino acids involved with inhibitor binding (Fig. 5c). The critical role of the conserved residues surrounding this binding site, such as Y195, Y208, Y335, F477, and F523, is supported by detailed studies of rat SQLE, where mutagenesis of equivalent amino acids resulted in loss of catalytic activity[28].

**Biochemical mechanism of action of SQLE inhibitors**. To further understand how NB-598 and Cmpd-4″ exert their effects, we undertook a more detailed biochemical dissection of their mechanism of action. Notably, both NB-598 and Cmpd-4″ exhibited time-dependent inhibition (Fig. 6a), which along with the high potency observed, suggested a slow tight-binding mode of inhibition. Further studies indicated non-competitive inhibition (Fig. 6b), despite crystallographic evidence suggesting that NB-598 and Cmpd-4″ were binding in the active site and would be expected to be competitive. While a previous report in fact described a competitive mechanism of action for NB-598[12], our biochemical studies suggest that, once bound in the active site, both NB-598 and Cmpd-4″ are more resistant to displacement by excess substrate. We hypothesize that the observed binding potencies are at the upper limit of the enzyme-inhibitor system and if higher squalene concentrations were feasible to achieve in the biochemical assay, it might be possible to competitively displace the inhibitors. In the absence of published progress curves in the previous work and similar substrate concentrations, it is difficult to fully account for the differences in the observed mechanisms of action. However, the structural insights, such as the domain motions and the Y195 conformational switch, are consistent with the observed tight-binding mechanism of inhibitor action, which can kinetically present as non-competitive inhibition, even upon binding in the active site at the same location as substrate[29,30].

**Implications for SQLE catalysis**. Among the FAD monooxygenases that catalyze a variety of diverse reactions, two-component epoxidation catalysis of Class E enzymes is among the least understood[31]. This class of epoxidases relies on external P450R for the transfer of reduced flavin, an initial step in the reaction. Binding of reduced flavin is followed by its reaction with oxygen to form a highly reactive intermediate C4a-(hydro)peroxyflavin at the flavin N5 position[32]. The best studied member of this family is styrene monooxygenase from *Pseudomonas putida*, but the reported structure lacks FAD, thus preventing further understanding of flavin stabilization and identification of catalytic residues[33]. Remarkably, analysis of our structures near the flavin N5-C4a atoms, a critical region that controls oxygen transfer, revealed that a side chain hydroxyl group from a conserved Y335 interacting with N5 via a bridging water molecule that is further anchored to the protein by main chain hydrogen bonds to I162 and E165 (Fig. 7). The volume available above the plane of flavin would indicate a face-on oxygenation and C4a-(hydro) peroxyflavin intermediate generation. Our model suggests that this Y335 water-bridged interaction may play a critical role in recruitment of reduced flavin and stabilization of this reactive intermediate to maintain catalytic efficiency. Indeed, Y334A mutation of equivalent residue in rat SQLE showed a significant decrease in catalytic efficiency[28]. The orientation of squalene from our docked model and the placement of the 2,3-vinyl group in a plane above the flavin ring (Fig. 7) is consistent with the requirement for stereo- and regio-specific catalysis for the formation of the product 2,3(*S*)-oxidosqualene.

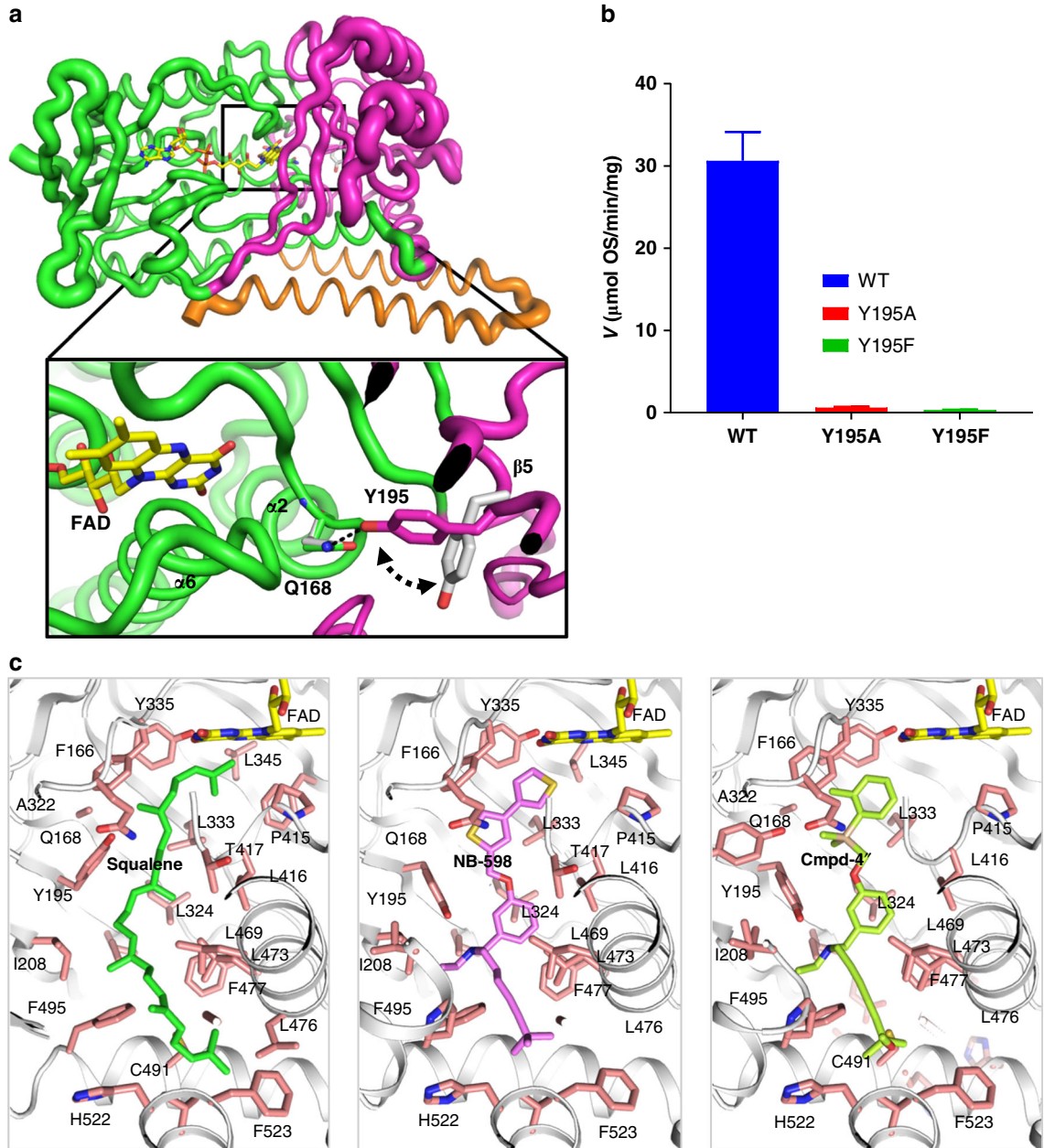

**Fig. 5** SQLE conformational rearrangements and the role of Y195. **a** Domain rearrangements in the unliganded SQLE structure in comparison with inhibitor-bound structure. Overall structure is shown in putty tube representation with the scaling to reflect the RMSD value calculated between SQLE•FAD and SQLE•FAD•Cmpd-4'' structures. FAD-binding domain in green, the substrate-binding domain in magenta, and the C-terminal membrane–associated helical domain in orange. FAD is in stick presentation with carbon atoms in yellow. A relatively higher degree of changes is observed for substrate-binding domain in comparison with other domains. The conformational change for Y195 between SQLE•FAD (in magenta) and SQLE•FAD•Cmpd-4'' (in gray) is illustrated in the inset with dashed black arrows. The hydrogen bond between Y195-Q168 in the unliganded SQLE•FAD structure is indicated by black dashed line. **b** Y195 mutants abrogate SQLE specific activity. SQLE enzymatic activity was assessed using the baculosome overexpression system. Bars represent the mean and standard deviation of triplicate measurements. **c** Side by side comparison of substrate-bound model of SQLE (right) with SQLE•FAD•NB-598 (center) and SQLE•FAD•Cmpd-4'' (left) structures. Squalene was docked into the long extended non-polar pocket of the unliganded structure within the substrate-binding domain. Side chains of amino acids in the binding site (light pink), FAD (yellow), squalene (green), NB-598 (pink), and Cmpd-4'' (light green) molecules are in stick representation and shown in the same relative orientation

## Discussion

In summary, we report the de novo structure of SQLE catalytic domain that is captured with a bound FAD cofactor and potent inhibitors. Our analysis of the unliganded structure provides an understanding of the obligate conformational rearrangements required for inhibitor binding and, as such, lays the foundation for further development of the next generation of SQLE inhibitors. Furthermore, our studies provide a structural explanation for the large therapeutic window observed in the clinical use of terbinafine and enable the interpretation of terbinafine-resistant

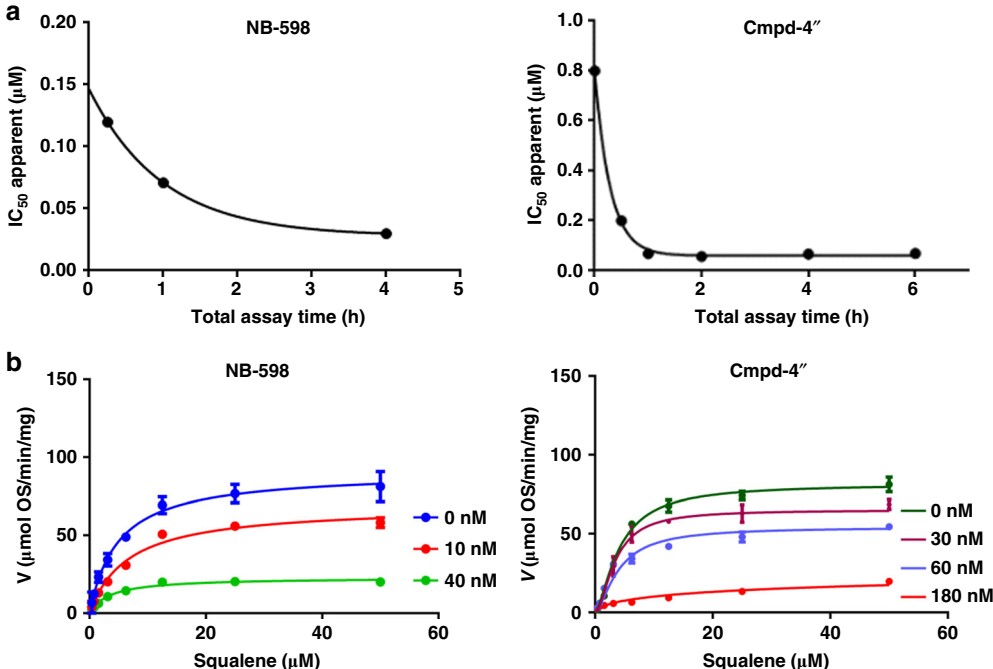

**Fig. 6** Mechanism of action of SQLE inhibitors. **a** Time-dependent inhibition potency of NB-598 and Cmpd-4′′. The $IC_{50}$ of NB-598 decreases from 120 nM to 30 nM and that of Cmpd-4′′ decreases from 800 to 69 nM upon increasing incubation time. **b** Mechanism-of-action study of NB-598 and Cmpd-4′′. NB-598 study was performed with fixed concentrations of inhibitor at 0, 10, and 40 nM indicating non-competitive inhibition with respect to squalene substrate. Cmpd-4′′ study was performed with fixed concentrations of inhibitor at 0, 30, 60, and 180 nM indicating non-competitive inhibition with respect to squalene substrate. Points and error bars are the mean and standard deviation of triplicate experiments

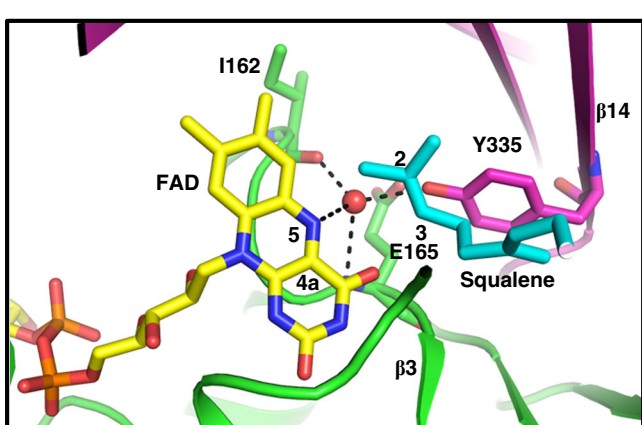

**Fig. 7** FAD stabilization and implications for catalysis. FAD stabilization by Y335 water–bridged interaction. Hydrogen bonds to flavin N5 position to water and its network to Y335, I162, and E165 are illustrated by dashed lines with water shown as a red sphere. FAD (from SQLE•FAD•Cmpd-4′′ complex) and docked, superposed squalene (in cyan) are in stick representation with atom positions 2 and 3 shown as sites of stereo- and regio-specific epoxidation

mutations. More broadly, substrate-binding model presented here sheds light on the catalytic mechanism of this class of monooxygenases.

## Methods

**Expression and purification of recombinant SQLE protein.** The cDNA of human SQLE (NCBI accession number: NM_003129.3) was cloned into pET28a expression vector (Supplementary Table 2). Multiple constructs were designed with six unique N-termini (starting at 1, 101, 111, 118, 124, 144) and four C-termini

(ending at 488, 518, 543, 574), along with different affinity/solubilization tags expressed in Rosetta (DE3) *E. coli* (Shanghai Weidi Biotechnology). Soluble proteins were obtained for four of the expression constructs with N- and C-terminal boundaries corresponding to 101–574, 118–574, 101–488, and 118–488 and were used in crystallization trials. The $His_6$-MBP-TEV-SQLE (118–574) construct in pET28a vector yielded crystals for structure determination. A detergent screen performed using thermal shift assay led to the identification of 3-[(3-cholamidopropyl)dimethylammonio]-1-propanesulfonate (CHAPS) as having the most stabilizing effect. CHAPS detergent was subsequently used for all extraction, purification and crystallization studies of SQLE protein.

The plasmid bearing the sequence encoding SQLE was transformed into Rosetta (DE3) *E. coli*. Cells were grown at 37 °C and induced with 1 mM isopropyl β-D-1-thiogalactopyranoside for about 20 h at 15 °C in Luria Broth media. SeMet incorporated SQLE (118–574) was expressed in M9 media supplemented with glucose, vitamins, and amino acids with L-methionine substituted by SeMet. *E. coli* cell pellets were harvested and resuspended in buffer A (50 mM Tris, 500 mM NaCl, 20 mM imidazole, pH 8.0, 0.5% CHAPS), and lysed two times by using a Microfluidizer (Microfluidics Corp, USA) at 15,000 psi and then subjected to ultracentrifugation at 40,000× *g* for 1 h. The supernatant was first loaded on a $Ni^{2+}$ Sepharose FF column (GE Healthcare, USA). The column was washed with buffer A and eluted with buffer B (50 mM Tris, 500 mM NaCl, 500 mM imidazole, pH 8.0, 0.5% CHAPS). Pooled protein sample was desalted with HiPrep 26/10 desalting 1 × 53 ml (Sephadex G-25F, GE Healthcare, USA) preequilibrated with buffer A. $His_6$-MBP tag was removed with TEV protease enzyme (*w*/*w*, protease to protein 1:40 ratio) by incubation at 4 °C overnight. Protein sample was recovered from the flow-through of the HisTrap HP column (GE Healthcare, USA) with buffer A, pooled and column desalted with buffer C (50 mM Tris, 50 mM NaCl, pH 8.0, 0.5% CHAPS). The sample was then purified with a Mono Q column by linear elution with Buffer D (50 mM Tris, 500 mM NaCl, pH 8.0, 0.5% CHAPS). The eluted sample was concentrated and further purified by size-exclusion chromatography using a HiLoad 16/600 Superdex 200 pg column (GE Healthcare, USA) equilibrated with Buffer D. Peak fractions were pooled and concentrated to about 8 mg/mL, flash-frozen in liquid nitrogen and stored at −80 °C for further use. The SeMet-derivatized SQLE protein sample was purified as described above, except for further purification by SEC. Protein was determined to be > 95% pure by SDS-PAGE and eluted as a monomer as assessed by SEC. Accurate-Mass Time-of-Flight (TOF) LC-MS (Agilent 6224 TOF LC-MS, USA) analysis was used to confirm the molecular weights of native and SeMet substituted proteins.

**Crystallization and X-ray diffraction data collection.** The random microseed matrix-screening[34] was performed at 20 °C by the sitting-drop vapor-diffusion

method using seeds of lysozyme crystals for the initial screening. Purified SQLE (118–574) at a concentration of 8 mg/mL was incubated with a 5-fold molar excess of Cmpd-4″ or NB-598 at 4 °C overnight. Optimal crystals could be grown by mixing 200 nanoliter (nL) protein•inhibitor complex solution with 180 nL reservoir solution consisting of 0.2 M ammonium sulfate, 0.1 M tri-sodium citrate pH 5.6, 15–18 % (w/v) PEG 4000, 20 nL seeding solution, and 40 nL 0.1 M hexammine cobalt(III) chloride as an additive and equilibrated against 15 μL of reservoir solution at 20 °C. Seeds were prepared by crushing either a large crystal of lysozyme or that of SQLE in about 60 μL well solution using a glass bead. Crystals appeared in 2–3 days and grew to a dimension of 50–150 μm in about a week. Mature crystals were harvested and cryo-protected in reservoir solution supplemented with 25% (v/v) glycerol and flash-frozen in liquid nitrogen. Crystals of SeMet-derivatized SQLE with Cmpd-4″, as well as unliganded SQLE were grown in the condition described above. All diffraction data were collected using single crystals at the Shanghai Synchrotron Radiation Facility on beamline BL17U1 equipped with an ADSC Q315r CCD detector 100 K and processed with HKL2000[35] and iMOSFLM[36]. Further processing was performed using the programs from the CCP4 suite.

**Structure determination and refinement**. For the SQLE•FAD•Cmpd-4″ structure determination, phases were derived from multiwavelength anomalous dispersion experiment performed at X-ray wavelength (λ) corresponding to selenium K-absorption peak (0.97909 Å), remote (0.96108 Å), and inflection (0.97959 Å) points and using PHENIX Autosol[37]. Initial phases were improved by manual model building and refinement performed with the native 2.30 Å dataset (collected at λ = 0.97941 Å) using COOT[38], PHENIX, and REFMAC[39]. The Ramachandran analysis for this structure showed 97.32% favored, 2.46% allowed, and 0.22% outliers. Structure of SQLE•FAD•NB-598 (data collected at λ = 0.97915 Å) was determined using the phases derived from SQLE•FAD•Cmpd-4″ co-ordinates, performing difference Fourier calculations, and iterative model building and refinement with the Ramachandran analysis on final model showing 97.54% favored, 2.24% allowed, and 0.22% outliers. Structure of unliganded SQLE•FAD (data collected at 0.97920 Å) was determined by molecular replacement with PHASER using SQLE•FAD•Cmpd-4″ co-ordinates as the starting model, iterative model building and refinement with the Ramachandran analysis on the final model showing 94.20% favored, 5.69% allowed and 0.11% outliers. The data collection and structure refinement statistics are summarized in Table 2.

**Structure analysis**. All figures representing structures were prepared with either PyMOL (https://www.pymol.org) or MOE (https://www.chemcomp.com/). Multiple sequence alignments were performed using CLUSTAL W and secondary structure mapping to sequence alignment using ESPript server (http://espript.ibcp.fr)[40]. Docking simulations and computational analysis were performed using MOE.

**Thermal shift assay**. Thermal shift assays were performed in 96-well PCR plates using an RT-PCR instrument (ABI 7500 Fast, Applied Biosystems, USA). Standard assay conditions (20 μL) contain 12.5 μL protein (1 mg/mL) in storage buffer 50 mM Tris (pH 8.0), 500 mM NaCl, 5% glycerol and 100 μM corresponding ligands, 5 μL Protein Thermal Shift™ buffer, and 2.5 μL of the ROX dye (8 ×, Protein Thermal Shift™ Dye Kit, Applied Biosystems, USA). All experiments were performed in triplicate and reaction mixtures were pre-incubated at 4 °C for 1 h. The fluorescence was measured at 1 °C temperature intervals from 25 °C to 99 °C. For the detergent screening, the SQLE protein was buffer exchanged to 500 mM NaCl, 50 mM Tris-HCl (pH 8.0) containing different detergents using Superdex™ 200 Increase 10/300 GL gel-filtration column (GE Healthcare, USA) and assayed as described above.

**Insect cell expression and preparation of baculosomes**. pFastBac1-SQLE was transformed into DH10Bac E. coli (Thermo Fisher) for transposition into bacmid, and positive clones were selected on a blue/white LB agar plate. Recombinant bacmid as isolated from positive clones with the PureLink Hipure Plasmid DNA miniprep kit (Thermo Fisher) and transfected into Sf9 cells (Thermo Fisher) to generate recombinant virus stocks, which were amplified further before Sf9 infection at a multiplicity of infection of two. Infected cells were harvested 72 h post infection and resuspended in phosphate buffered saline, then disrupted by sonication. The cell lysate was clarified by centrifugation at 1500× g for 10 min, the supernatant was further centrifuged at 10,000× g for another 10 min to remove cell debris; and then the supernatant was harvested and centrifuged at 100,000× g for 90 min. The pellet of SQLE baculosomes was collected and homogenized in the buffer (50 mM potassium phosphate, 20% Glycerol, pH7.0) by micro-electric homogenizer (IKA Works, Germany). The protein concentration was determined by the Bradford method. SQLE baculosomes were aliquoted and stored at −80 °C. The abundance of SQLE in the baculosomes compared to other subcellular fractions was evaluated by immunoblotting against the N-terminal his-tag. Y195A, and Y195F mutant baculosomes were prepared as described above from Sf9 cells infected with engineered viruses.

**Biochemistry assay of SQLE activity**. All biochemistry assay buffers contained 100 mM Tris, 0.1% triton X-100, 1 mM EDTA, 0.1 mM FAD and 1 mM NADPH.

For IC$_{50}$ potency determination, compounds (50 μL) and SQLE baculosomes mixture (0.5 mg/mL final concentration, in 25 μL) were pre-incubated at 25 °C for 1 h, and then substrate squalene (100 μM final concentration, in 25 μL) was added to start the reaction at 37 °C in a water bath for 1 h. The reaction was quenched by adding 100 μL of 10 mM ammonium acetate buffer, followed by addition of 600 μL ethyl acetate containing 0.4 mg/mL butylated hydroxytoluene (BHT) to prevent oxidation. The mixture was vortexed for 30 min to extract 2,3-oxidosqualene, then centrifuged at 4000 rpm for 10 min. The supernatant was transferred to a new plate for LC-MS/MS detection. For mechanism of action studies, the final squalene concentration was varied from 50 to 0.39 μM. For the time-dependent inhibition, compound incubation time was varied from 0 to 6 h and the reaction was reduced to 15 min. Kinetic analysis for HLMs (Sekisui Xenotech, LLC, Kansas City, KS) was performed as for baculosomes, with HLM concentration at 0.5 mg/ml. Kinetic analysis for recombinant SQLE was performed as for baculosomes, with the addition of 1.5 μg/ml recombinant P450R (Sigma-Aldrich). Data from the mechanism-of-action study were globally fit to the model for α = 1 non-competitive inhibition using the equation (1).

$$v = \frac{V_{max}[S]}{([S] + K_M)\left(1 + \frac{[I]}{K_i}\right)} \tag{1}$$

**LC-MS/MS analysis of SQLE reaction**. Samples were dried down under nitrogen and then reconstituted with 70 μL of 0.2 mg/mL BHT in acetonitrile. An aliquot of 7.5 μL was injected into the UPLC-MS/MS system. The instrument setup consisted of an AB Sciex API-6500 Mass Spectrometer (AB Sciex, USA) equipped with a Waters UPLC Acquity (Waters, USA). The UPLC separation was performed on an ACQUITY UPLC BEH C18 (2.1 × 50 mm, 1.7 μm, Waters) at 40 °C. Formic acid in water (0.1%, v/v, mobile phase A) and a mixture of acetonitrile and isopropanol (80:20, v/v with 0.1% formic acid, mobile phase B) were employed as the mobile phase. An isocratic elution of 98% mobile phase B was used and run time was 1.5 min. The flow rate of mobile phase was set at 0.6 mL/min. Squalene and 2,3-oxidosqualene were ionized under a positive ion spray mode and detected through the multiple-reaction monitoring of a mass transition pair at m/z 414.4 → 231.0 and 427.4 → 409.5, respectively.

**Reporting summary**. Further information on experimental design is available in the Nature Research Reporting Summary linked to this article.

## Data availability

The atomic co-ordinates and structure factors for all complex structures are deposited in the Protein Data Bank (https://www.rcsb.org) with deposition IDs 6C6N for SQLE•FAD•Cmpd-4″, 6C6P for SQLE•FAD•NB-598, and 6C6R for SQLE•FAD, which will be available upon publication. A reporting summary for this Article is available as a Supplementary Information file. Other data are available from the corresponding author upon reasonable request.

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

## Acknowledgements

We thank Professor Karen Allen of Boston University for helpful discussions and advice on preparation of this manuscript. We thank Zhijia Lv and Yaliang Huang for help with purification, crystallization, and thermal shift assays and Mingzong Li for assistance with the $^{13}$C NMR experiments. This work was supported by Agios Pharmaceuticals.

## Author contributions

Design, planning, structure determination, refinement, and data analysis: A.K.P., S.G., and L.J.; docking analysis: G.C.; cloning, crystallization, data collection, structure determination, and refinement: F.W. and R.W.; cloning and protein purification C.F.; biochemical assays: X.L. and R.N.; design of compound synthesis: Z.S. and J.P.-M.; manuscript draft: A.K.P., S.G., L.J., G.C., J.P.-M., and G.A.S.; and manuscript editing: A.K.P., S.G., L.J., G.C., R.N., S.A.B., L.D., C.E.M., N.N., D.P., Z.S., J.P.-M., and G.A.S.

## Additional information

**Competing interests:** A.K.P., S.G., R.N., S.A.B., L.D., C.E.M., N.N., D.P., and Z.S. are employees of and have ownership interest in Agios Pharmaceuticals. The remaining authors declare no competing interests.

