## [Peer Review File · Nature Communications]

Reviewers' comments:

Reviewer #1 (Remarks to the Author):

Padyana and colleagues report the first structure of a key rate-limiting enzyme in cholesterol synthesis, squalene epoxidase/monooxygenase. The catalytic domain is well conserved from fungi to humans, and the enzyme is the target of fungicides and of increasing interest in human health and disease (e.g. SQLE is implicated in a growing list of cancers). Here they solve the structure of human SQLE at a resolution of 2.3/2.5 Å, with and without two similar pharmacological inhibitors. They focus on elucidating the binding site and mechanism of action of these inhibitors. My comments below are designed to further strengthen this highly significant piece of work.

Major comments:

1. Catalytic domain: As reported previously [ref#11] the N-terminal domain clearly causes problems when trying to work with recombinant full-length protein. Hence, the crystal structures solved lacks the first 117 amino acids (~a fifth of the enzyme). Therefore, to be more precise the title should be changed to "Structure and inhibition mechanism of the catalytic domain of human squalene epoxidase".
2. Native substrate: Given the author's pharmaceutical background, their focus on inhibitors is understandable. However, from a more basic biology viewpoint, it would also be useful to have a fuller description/discussion of implications for the actions of SQLE on converting its substrate, squalene, to its product, 2,3-oxidosqualene. Molecular docking experiments are presented for squalene. Does this mean that crystallization experiments were not attempted for SQLE with squalene? The authors should clarify this. What are the implications for the Q168-Y195 switch for squalene epoxidation? E.g. Could this facilitate the epoxidation itself or perhaps product release? What is the overlap between the putative substrate binding pocket and the cavity occupied by the inhibitors? There appears to be considerable overlap whereas non-competitive inhibitors traditionally bind at an allosteric site – Please comment. Showing the squalene binding model side by side with the inhibitor binding with identical orientation/formatting would be useful. The authors should also mutate the conserved Y195 mutation to further strengthen evidence for the importance of this site by showing that mutation abolishes affinity of the inhibitors.
3. Yeast parallels: SQLE/ERG1 is of great interest as an anti-fungal target with extensive medical and agricultural applications. Some discussion of this would be useful. Notably, the yeast sequence should be added to Supp Fig. 4, and molecular docking simulations should also be done with the archetype ERG1 inhibitor, terbinafine. This compound shares the tertiary amine motif of the two inhibitors tested but has a naphthalene moiety which would be more compact than the corresponding moieties in either NB-598 or Cmpd-4". This is particularly timely because terbinafine is being increasingly talked about in terms of being repurposed for the treatment of human disease, particularly cancers. So how does terbinafine compare with NB-598/Cmpd-4" in terms of inhibitory potency for human SQLE and does your structure help explain this?

Other comments:

1. I can't find any reference to the source and protocol for the preparation of human liver microsomes. Please provide this, as well as the details of the appropriate Institutional Human Ethics clearance.
2. Page 2: SQLE is particularly significant as it is the rate-limiting enzyme in cholesterol synthesis after HMGCR. Please state this in your introduction. S Gill et al. Cell Metabolism 2011 would be an appropriate reference to cite.
3. Page 3, top: Please reference "have been previously reported"
4. Page 3, bottom: Please give errors for all values, and please state the HLM SQLE K_{cat} and K_{cat}/K_m explicitly in the text compared to SQLE (118 - 574).
5. Page 7: Please correct "where higher levels of squalene..."
6. Please consider including the resolution in the Abstract.

Reviewer #2 (Remarks to the Author):

The manuscript "Structure and inhibition mechanism of human squalene epoxidase" by Padyana et al. reports the first crystal structures of human SQLE with and without inhibitors. Evaluating several bacterial expression constructs, the authors use T_m shift assays to identify optimal boundaries for inhibitor co-crystallisation and demonstrate that the catalytic activity of the construct is somewhat comparable to full length human protein. The three resulting SQLE-FAD crystal structures with and without the previously reported SQLE inhibitors NB-598 and Cmpd-4" reveal the molecular architecture of SQLE and enable the authors to rationalise previously observed inhibitor SAR including a key hydrogen bond of the core tertiary amine scaffold with tyrosine 195. Based on these and previous observations the authors develop inhibitor analogues with close chemical similarity but strongly reduced activity to support further phenotypic studies. The authors also investigate the effect of the inhibitors on SQLE activity and present a hypothetical binding mode for squalene and its stereospecific conversion into oxidosqualene.

In general, the manuscript is well written and reveals new and important insights into the activity of a potential drug target. Data and procedures are mostly well described and therefore publication is recommended pending a few minor edits. For example, the main figures should be expanded to include some of the important biochemical data currently presented in the SI (e.g. SI Fig 1, 2, 7 (merge with SI Fig 6?).

- SI Fig 1a/b – please include original T_m shift curves.
- Information on compound purity should be added (method, estimated purity of sample); HPLC is mentioned in the general chemistry part but no details/results are included for compounds. For novel compounds ¹³C data should be added.
- The authors present some interesting data on inhibitor kinetics/residence time which could be complemented by comparison with SPR results

Reviewers' Comments and author responses:

Reviewer #1 (Remarks to the Author):

Reviewer comment 1-0:

Padyana and colleagues report the first structure of a key rate-limiting enzyme in cholesterol synthesis, squalene epoxidase/monooxygenase. The catalytic domain is well conserved from fungi to humans, and the enzyme is the target of fungicides and of increasing interest in human health and disease (e.g. SQLE is implicated in a growing list of cancers). Here they solve the structure of human SQLE at a resolution of 2.3/2.5 Å, with and without two similar pharmacological inhibitors. They focus on elucidating the binding site and mechanism of action of these inhibitors. My comments below are designed to further strengthen this highly significant piece of work.

We thank the Reviewer for their time and recognition of the significance of this work. We appreciate the constructive suggestions provided to strengthen the manuscript.

Reviewer comment 1-1:

1. Catalytic domain: As reported previously [ref#11] the N-terminal domain clearly causes problems when trying to work with recombinant full-length protein. Hence, the crystal structures solved lacks the first 117 amino acids (~a fifth of the enzyme). Therefore, to be more precise the title should be changed to "Structure and inhibition mechanism of the catalytic domain of human squalene epoxidase".

We agree with the Reviewer's comment. We have changed the title to "Structure and inhibition mechanism of the catalytic domain of human squalene epoxidase".

Reviewer comment 1-2:

2. Native substrate: Given the author's pharmaceutical background, their focus on inhibitors is understandable. However, from a more basic biology viewpoint, it would also be useful to have a fuller description/discussion of implications for the actions of SQLE on converting its substrate, squalene, to its product, 2,3-oxidosqualene. Molecular docking experiments are presented for squalene. Does this mean that crystallization experiments were not attempted for SQLE with squalene? The authors should clarify this.

We appreciate the feedback regarding our discussion on native substrate and the observed mechanism of action of the inhibitors. We attempted to generate co-crystals of SQLE-FAD in complex with substrate (squalene) or product (oxidosqualene). However, the crystals obtained did not produce diffraction data and we clarify this point now in the updated manuscript.

Reviewer comment 1-3:

What are the implications for the Q168-Y195 switch for squalene epoxidation? E.g. Could this facilitate the epoxidation itself or perhaps product release?

Please see below for our discussion of the Y195 mutants.

Reviewer comment 1-4:

What is the overlap between the putative substrate binding pocket and the cavity occupied by the inhibitors? There appears to be considerable overlap whereas non-competitive inhibitors traditionally bind at an allosteric site – Please comment.

This is a great suggestion and we have now included this information as Figure 5c. The associated text clearly explains the significant overlap between putative substrate binding pocket in the cavity occupied by the inhibitors.

We have also added further discussion in the manuscript on how these inhibitors (exhibiting slow-tight binding) appear as non-competitive in kinetic studies due to their high affinity, insufficient substrate concentration in the assay due to technical reasons, and the Y195 conformational rearrangements. Our discussion now also includes references to other published systems that display conceptually similar results.

Reviewer comment 1-5:

Showing the squalene binding model side by side with the inhibitor binding with identical orientation/formatting would be useful.

As suggested by the Reviewer, we have now included a side by side comparison of squalene binding model with both inhibitors (NB-598 and Cmpd-4") with identical orientation and formatting as Figure 5c.

Reviewer comment 1-6:

The authors should also mutate the conserved Y195 mutation to further strengthen evidence for the importance of this site by showing that mutation abolishes affinity of the inhibitors.

As suggested by the Reviewer, we have conducted a series of mutagenesis experiments to characterize the role of Y195. We engineered two amino acid substitutions, either Y195F or Y195A, in the full-length SQLE and studied their impact on the biochemical activity in the baculosome assay. Both mutations resulted in a >90% loss of catalytic activity strengthening the hypothesis that hydrogen bonding property of side-chain hydroxyl of Y195 to Q168 side chain is critical for maintaining the SQLE activity. However due to loss of the intrinsic enzymatic activity for both of the mutations, we were unable to test their contribution to inhibition mediated by NB-598 or Cmpd-4". We now present this data in Figure 5b and discuss the implications in the results and discussions sections.

Reviewer comment 1-7:

3. Yeast parallels: SQLE/ERG1 is of great interest as an anti-fungal target with extensive medical and agricultural applications. Some discussion of this would be useful. Notably, the yeast sequence should be added to Supp Fig. 4, and molecular docking simulations should also be done with the archetype ERG1 inhibitor, terbinafine. This compound shares the tertiary amine motif of the two inhibitors tested but has a naphthalene moiety which would be more compact than the corresponding moieties in either NB-598 or Cmpd-4". This is particularly timely because terbinafine is being increasingly talked about in terms of being repurposed for the treatment of human disease, particularly cancers. So how does terbinafine compare with NB-598/Cmpd-4" in terms of inhibitory potency for human SQLE and does your structure help explain this?

We would like to thank the reviewer for this constructive suggestion. Recognizing the importance of the role of anti-fungal SQLE inhibitors as therapeutic agents and the use of terbinafine as a tool in human cells, we conducted several additional studies described below.

- 1) We have biochemically tested terbinafine against human SQLE using our HLM assay. We find it to be a weak partial inhibitor with a relative IC₅₀ of 7.7 μM and a maximal inhibition of 65% at 100 μM inhibitor concentration. Our studies suggest that NB-598 or Cmpd-4" are superior tools for studying human SQLE biology, as compared to terbinafine, particularly in the context of preclinical studies. We present this data in Figure 3a and 3b.
- 2) We have added four fungal SQLE sequences to our sequence alignment figure: three pathogenic organisms *Trichophyton rubrum*, *Trichophyton mentagrophytes*, *Candida albicans*, and non-pathogenic *Saccharomyces cerevisiae*. This is now provided in the updated Supplementary Fig 3a.
- 3) Terbinafine is structurally similar to NB-598 and the highly conserved active site of fungal and mammalian SQLE allows us to model the structure of terbinafine in the binding site of NB-598. We found that multiple amino acids positioned near the aromatic side of the inhibitor were not conserved between the species (F166, I197, and L324), while the amino acids near the aliphatic side were identical between human and fungal SQLE. The aromatic side of terbinafine contains bulkier naphthalene group in the position of benzene linker of NB-598. Modeling the terbinafine using NB-598 template in human SQLE positions the naphthalene group adjacent to bulkier hydrophobic side chains of I197 and L324. These sub-optimal non-polar contacts are consistent with the observed higher IC₅₀ values of terbinafine in the HLM enzymatic assay. Interestingly, residues corresponding to I197 and L324 in dermatophyte SQLE are smaller hydrophobic valines, likely resulting in optimal interactions with naphthalene consistent with the reported selectivity profile of terbinafine. These results are now shown in Figure 3c and are further discussed in the text of the updated manuscript.
- 4) Several reports have identified strains resistant to terbinafine treatment with point mutations detected in fungal SQLE (*ERG1* gene) in both clinical and non-clinical settings. We mapped the reported resistant point mutations onto the human SQLE sequence and to the SQLE•FAD•NB-598 structure. Remarkably, all the SQLE resistant mutations are in the inhibitor binding pocket. Mutation of these conserved residues (L326, L473, F477, F492, F495, L508, P505, and H522 of human SQLE) would be predicted to affect the non-polar interactions with the inhibitor resulting in the loss of biochemical potency. The resistance mutations are now summarized in

Supplementary Table 1. The location of individual resistance mutations is illustrated on Figure 3d.

Collectively, our structural insights are consistent with the biochemical results and provide a detailed explanation for the weak inhibitory potency of terbinafine against human SQLE and offer new understanding of the previously identified terbinafine-resistant mutations in fungal SQLE. We updated the manuscript in multiple places (Abstract, Results, and Discussion) to appropriately integrate this information.

Reviewer comment 1-8:

1. I can't find any reference to the source and protocol for the preparation of human liver microsomes. Please provide this, as well as the details of the appropriate Institutional Human Ethics clearance.

Human liver microsomes were purchased from a commercial vendor. We have now added the commercial source (Sekisui Xenotech, LLC, Kansas City, KS) in the methods section for *Biochemistry assay of SQLE activity*. Institutional Human Ethics clearance is not required for the purchase and use of this commercially-available reagent.

Reviewer comment 1-9:

2. Page 2: SQLE is particularly significant as it is the rate-limiting enzyme in cholesterol synthesis after HMGCR. Please state this in your introduction. S Gill et al. Cell Metabolism 2011 would be an appropriate reference to cite.

We thank Reviewer for this constructive feedback. We now include this reference in describing multiple mechanisms of SQLE regulation and updated the introduction section.

Reviewer comment 1-10:

3. Page 3, top: Please reference "have been previously reported"

We thank the reviewer for identifying this error. We have now moved the two cited references appropriately to position within this sentence and captured in the reference # 9 and 10.

Reviewer comment 1-11:

4. Page 3, bottom: Please give errors for all values, and please state the HLM SQLE K_{cat} and K_{cat}/K_M explicitly in the text compared to SQLE (118 - 574).

We have now added error values for all the enzymatic parameters displayed and state the HLM SQLE k_{cat} and k_{cat}/K_M explicitly in the text compared to SQLE (118 - 574). To facilitate the clarity of presentation, this information is now displayed as a table in Figure 1d.

Reviewer comment 1-12:

5. Page 7: Please correct “where higher levels of squalene...”

We apologize for this typographical error. It has been fixed.

Reviewer comment 1-13:

6. Please consider including the resolution in the Abstract.

We have added the resolution to all the individual crystal structures mentioned in the abstract.

Reviewer #2

Reviewer comment 2-0:

The manuscript “Structure and inhibition mechanism of human squalene epoxidase” by Padyana et al. reports the first crystal structures of human SQLE with and without inhibitors. Evaluating several bacterial expression constructs, the authors use Tm shift assays to identify optimal boundaries for inhibitor co-crystallisation and demonstrate that the catalytic activity of the construct is somewhat comparable to full length human protein. The three resulting SQLE-FAD crystal structures with and without the previously reported SQLE inhibitors NB-598 and Cmpd-4” reveal the molecular architecture of SQLE and enable the authors to rationalise previously observed inhibitor SAR including a key hydrogen bond of the core tertiary amine scaffold with tyrosine 195. Based on these and previous observations the authors develop inhibitor analogues with close chemical similarity but strongly reduced activity to support further phenotypic studies. The authors also investigate the effect of the inhibitors on SQLE activity and present a hypothetical binding mode for squalene and its stereospecific conversion into oxidosqualene.

In general, the manuscript is well written and reveals new and important insights into the activity of a potential drug target. Data and procedures are mostly well described and therefore publication is recommended pending a few minor edits. For example, the main figures should be expanded to include some of the important biochemical data currently presented in the SI (e.g. SI Fig 1, 2, 7 (merge with SI Fig 6?).

We appreciate Reviewer’s enthusiasm about our work. We appreciate the constructive suggestions provided to strengthen the manuscript. As suggested by the reviewer, we have now moved the figures displaying the biochemical data from supplementary materials into the main section. The biochemical data is now displayed in **Figure 1c, Figure 1d, Figure 4, and Figure 6**.

Reviewer comment 2-1:

- SI Fig 1a/b – please include original Tm shift curves.

We have now included the original Tm shift curves for the supplementary figures as displayed in **Supplementary Fig 1 (a) and Supplementary Fig 1 (b)**

Reviewer comment 2-2:

- Information on compound purity should be added (method, estimated purity of sample); HPLC is mentioned in the general chemistry part but no details/results are included for compounds. For novel compounds 13C data should be added.

We have now added the HPLC purity information to the supplementary data provided for the compound synthesis.

For novel compounds (NB-598.ia and Cmpd-4".ia) we have now collected 13C data and included this in the supplementary data.

Reviewer comment 2-3:

- The authors present some interesting data on inhibitor kinetics/residence time which could be complemented by comparison with SPR results.

We agree with the Reviewer that additional biophysical methods, such as SPR, to characterize the mechanism of action of inhibitors would be informative. However, the development of these additional assays is neither easy nor fast, and thus is outside of the scope of this initial manuscript. It is important to recognize that such assays will be important in future studies aimed at the development of the next generation of SQLE inhibitors.

REVIEWERS' COMMENTS:

Reviewer #1 (Remarks to the Author):

I'm very happy with how the authors have taken my comments on board, and think the manuscript has been greatly strengthened as a result.